# Banning new gas boilers as a no-regret mitigation option

Célia Escribe [1,2,4] ✉ & Lucas Vivier [1,3,4] ✉

The low uptake of low-carbon heating systems across Europe has prompted authorities to consider more ambitious measures, including a complete ban on the installation of new fossil fuel boilers. In this analysis, we simulate the impacts of introducing this ban in France under 11,664 scenarios covering major uncertainties. We find that the ban induces major changes in the energy system, leading to efficiency gains. Additionally, we find that the ban increases the likelihood of reaching carbon neutrality while reducing total system cost in over 75% of scenarios. Finally, we show that the implementation of the ban, when coupled with the existing subsidy framework, mitigates inequalities among owner-occupied households but generates adverse effects for those in privately rented homes.

Achieving carbon neutrality in the European residential sector requires a major switch from fossil fuel boilers to low-carbon energy sources such as electricity, solid biomass, or district heating[1]. In Europe, residential space heating represents 17% of total final energy consumption, with ~ 75% still relying on fossil fuels[2]. A major obstacle to the transition to low-carbon heating systems is that the social cost of carbon is typically not included in residential energy prices, so homeowners' investments are not aligned with environmental goals. In addition, homeowner behavior may deviate from the perfectly rational consumer assumed in standard microeconomic models, leading to suboptimal levels of investment. In particular, homeowners tend to undervalue future energy benefits[3] or express a bias for the existing technology[4] when making heating system investment decisions. Without proper policy instruments, these behaviors could drive excessive gas demand in the residential sector, hindering the achievement of climate targets. Environmental externalities and heterogeneous behavioral anomalies in the residential sector imply that the first-best policy mix should be a two-part instrument including perfectly targeted subsidies and a carbon tax[5]. It is, however, challenging to implement a policy mix that comes close to this optimum: carbon taxes at the socially optimal level are often politically unfeasible[6], and realistic subsidy designs cannot be individually targeted. Consequently, and despite efforts to implement market-based instruments[7], uptake rates of low-carbon heating systems across Europe remain low[8], leading authorities to consider more ambitious

measures. The uncertain nature of most of the parameters driving investment decisions in the residential and energy sectors increases the risk of misaligned price incentives. Such misalignment may result in unmet climate targets if subsidies are insufficiently ambitious, or distributional issues among households- between those receiving subsidies and those bearing the costs-if subsidies are high. In addition, the long heating systems lifetimes require a complex intertemporal approach to instrument design to avoid lock-in effects. Given these difficulties, a ban on fossil fuel boilers emerges as a pragmatic policy choice that makes it easier to achieve climate targets without having to rely on excessive subsidies.

Although several European Union (EU) Member States have already introduced ban measures to phase out fossil fuel boilers, these regulations affect only a minor share of the EU's heating energy consumption[9]. They mostly target new buildings with specific fuels like oil or include numerous exemptions. Therefore, the EU Commission has proposed to extend the ban to all standalone fossil fuel boilers across the EU from 2029, as per the EU Save Energy Plan[10]. Furthermore, the recent adoption of the Energy Performance Building Directives mandates that Member States implement measures to completely phase out fossil fuel heating and cooling by 2040[11]. In this context, EU Member States are currently considering implementing a complete ban on installing new fossil fuel boilers.

Economists often argue that regulatory instruments are less cost-effective than price-based policies. These policies fail to account for

[1]CIRED-CNRS, Nogent sur Marne, France. [2]CMAP, CNRS, Ecole Polytechnique, Institut Polytechnique de Paris, Palaiseau, France. [3]ENPC, Ecole des Ponts ParisTech, Champs-sur-Marne, France. [4]These authors contributed equally: Célia Escribe, Lucas Vivier. ✉e-mail: celia.escribe@polytechnique.edu; lucas.vivier@enpc.fr

the heterogeneity of households by imposing uniform requirements that may not be consistent with individual cost-effectiveness[12]. A major concern of the ban on gas boilers is the induced energy system externalities. Specifically, a rapid increase in space heating electricity demand concentrated during peak load could require further investments in the electricity sector, increasing overall costs and hampering the ability to achieve carbon neutrality. Little engineering research investigates how a large roll-out on heat pumps impacts the electricity system[13–16], but none considers the dynamics associated with a ban on gas boilers. In addition, these studies do not explore the cost-effectiveness and fairness of this ban. Despite the potentially massive impact and this controversial position, this measure has been little studied.

The objective of this paper is to assess the impact of implementing a ban on gas boilers in the residential sector. We address the following questions: To what extent does the ban contribute to achieving carbon neutrality, and what are its impacts on the energy system, total system costs, and distributional effects?

To answer these questions, we extend a modeling framework that integrates detailed bottom-up models for the energy and residential sectors[17]. The framework relies on two key features to assess the ban on fossil fuel boilers. First, the model simulates endogenous investments in home insulation and heating systems. Each homeowner upgrades their heating system or insulates their home based on a discrete choice model influenced by existing policies and market barriers such as credit constraints, behavioral anomalies, and hidden costs of energy-efficient technologies. This model is therefore suitable for comparing the effects of a ban, which is represented as a restriction of homeowners' choice set, with a current policy scenario that mimics implemented policies in France[18]. The policy mix includes subsidies for home insulation and low-carbon heating systems as well as a residential carbon tax of 45 euros per $tCO_2$ (€per $tCO_2$). Second, the model includes the main interactions between the residential sector and the energy system. The hourly resolution finely captures the impact of additional residential electricity demand on peak power load and the resulting investment needs in the electricity sector. In addition, the energy model allocates gas production to both residential gas boilers and the use of peaking power plants in the electricity sector. Low-carbon gas is produced either by biogas with its limited supply or by power-to-gas technologies, which in turn increase electricity demand. Consequently, our framework captures significant cross-sectoral interactions between residential and energy sectors, as well as between the two main energy vectors: gas and electricity. Finally, the model is open-source[19].

Taking France as a case study, we examine how the implementation of a ban on gas boilers - which is synonymous with a ban on all fossil fuels in France, as a ban on oil boilers has already been enacted - contributes to achieving carbon neutrality in the long term. To this end, we systematically compare two policy scenarios: the current policy scenario and an alternative scenario that adds a ban on gas boilers to the current policies. All simulations are done under a carbon budget constraint. We simulate 11,664 scenarios (half with the ban and just as many only with the current policy mix), capturing the main uncertainties driving investment dynamics in the energy and residential sectors (see Table 1). These include uncertain renewable and biomass potential capacities[20,21], volatile natural gas prices, and uncertain electricity demand in other sectors. We also consider the level of policy ambition to be uncertain, as it has varied considerably between 2005 and 2024[18]. The response of households to price changes, which is represented here by an average price elasticity parameter, is difficult to estimate and is also considered uncertain. In addition, the future efficiency and cost of heat pumps span a wide range[22]. Lastly, the 2050 carbon budget for the energy and residential sectors hinges on uncertain carbon sinks and abatement in other sectors. We evaluate the ban in terms of its robustness in achieving the carbon neutrality

target under uncertainty, its cost-effectiveness, and its distributional effects among the large set of plausible future scenarios[23].

This study makes four contributions to understanding the impact of a gas boiler ban. First, we demonstrate that the additional electricity demand resulting from the implementation of the ban does not have any adverse effects on the electricity system. Instead, it leads to reduced primary energy requirements and improved capacity factors for power plants. Second, we demonstrate that the ban increases the likelihood of meeting climate targets, showing no adverse effect on the electricity system while hedging against the lower-than-expected biogas potential. Third, we find that while the cost implications of the ban are highly dependent on uncertainty factors, it reduces total system costs in 75% of the scenarios analyzed. Fourth, we show that the distributional impacts are highly sensitive to the subsidy design for heat pumps, requiring consideration of both income and occupation status.

## Results
### A ban addresses energy service demands more efficiently
Despite the increasing number of dwellings (see Fig. 1b), home insulation policies reduce overall energy demand (Fig. 1a). The ban on gas boilers shifts residential gas consumption primarily to electricity, due to the limited availability of wood and district heating. In particular, gas boilers are mostly replaced by heat pumps as the most cost-efficient available option. This shift results in a 75% increase in electricity demand, which is particularly pronounced in the cold months when space heating demand peaks and the technical efficiency of heat pumps is at its lowest due to low outside temperatures. Supplementary Fig. 4a illustrates that the ban could raise peak electricity demand by up to 10 Gigawatt (GW) in 2050 compared to the current policy scenario.

Banning gas boilers leads to significant transformations within the energy system by (i) reducing primary energy needs, and (ii) improving the capacity factors.

First, Fig. 2a shows that the system relies on less primary energy to deliver the same energy services. By 2050, the ban will reduce the primary energy requirements by 12 Terawatt hours (TWh). These shifts are driven by different strategies for allocating gas resources, which are constrained by climate targets and limited biogas potential. While low-carbon gas is currently used in gas boilers, it could be redirected to peaking power plants that support electric heating systems if the ban is enforced. Overall, as shown in Supplementary Fig. 2, we find that the combination of peaking power plants and heat pumps meets energy service demands more efficiently.

Second, meeting peak demand with peaking power plants eliminates the need for the combination of renewable capacity and battery storage as a flexibility solution. Specifically, Fig. 2b demonstrates that the ban avoids the installation of 12 GW of renewable capacity (offshore wind and solar photovoltaic) and 3 GW of battery storage while instead requiring an additional 12 GW of peaking plants (Supplementary Table 1). This reduction in renewable capacity leads to a more efficient use of nuclear power as a base-load generator, thereby increasing its capacity factor.

### A ban is critical to meet carbon neutrality under uncertainty
We conduct simulations across 11,664 scenarios and find that 99% of these scenarios achieve carbon neutrality with the ban in place, compared to only 52% in the current policy scenario. On the one hand, scenarios that achieve carbon neutrality without the ban also succeed under the ban, indicating no adverse effects from its implementation. This suggests that the electricity system can dynamically and effectively adapt to the additional peak load, responding quickly to the heat pump roll-out induced by the ban. On the other hand, we show that incentives in the current policy package are not well-aligned with climate targets when considering a wide range of plausible futures.

**Table 1 | Uncertainty scenarios for model parameters**

| Parameter | Description | Values |
|---|---|---|
| **Energy system** | | |
| Biogas potential | Available potential for methanization and pyrogazification | Low, Reference*, High |
| Renewable potential | Available potential for solar pv, onshore and offshore wind | Low, Reference*, High |
| Gas prices | The growth rate for wholesale natural gas prices | Low, Reference*, High |
| **Residential Demand** | | |
| Technical progress heat pumps | How much will cost decrease in 2035 compared to 2018 ? | Low, Reference*, High |
| Insulation policy | Whether the policy package includes ambitious insulation policy | No, Yes* |
| Heater policy | Whether the policy package includes ambitious heater policy | No, Yes* |
| Heat pump price elasticity | Parameter driving households' heat pump price elasticity | Low, Reference*, High |
| **Global parameters** | | |
| Other electricity demand | Level of electricity demand for all sectors excluding residential space heating | Low, Reference*, High |
| Carbon budget | The trajectory of the available carbon budget for the residential and electricity sector | Low, Reference* |

*Corresponds to the value used in the reference configuration.

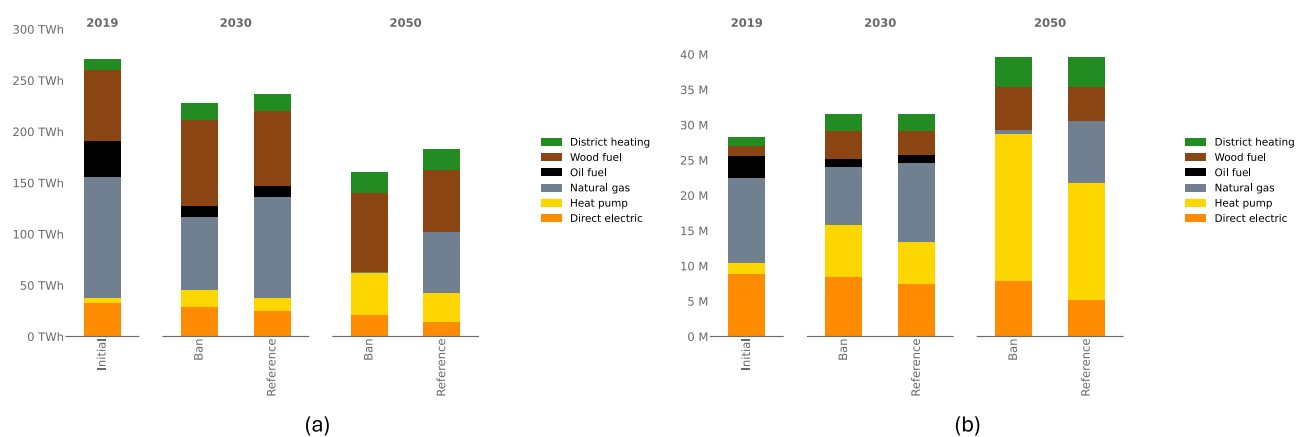

(a) (b)

**Fig. 1 | Evolution of heating system stock under policy scenarios. a** Space heating consumption in the residential sector (TWh per year). **b** Heating system stock (Million). The notation "Natural gas" corresponds to households heating their dwelling with gas boilers. Such heating systems may rely on renewable gas in addition to fossil gas. Source data are provided as a Source Data file.

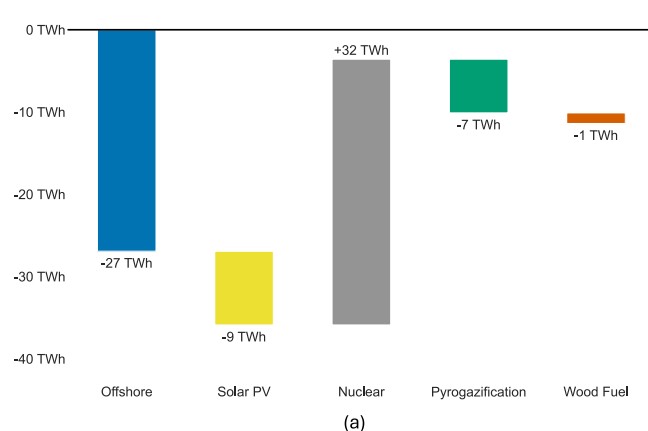

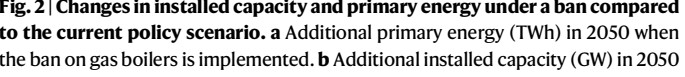

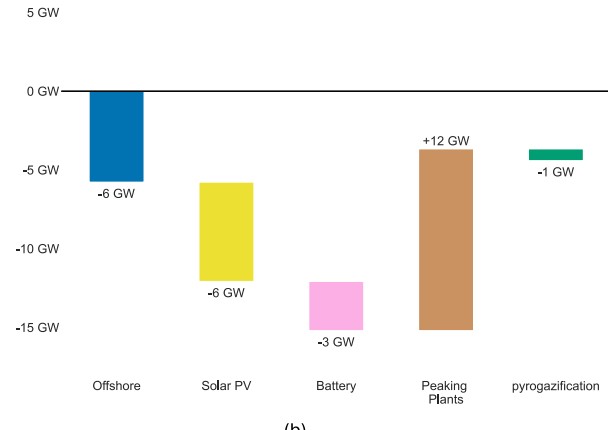

(a) (b)

**Fig. 2 | Changes in installed capacity and primary energy under a ban compared to the current policy scenario. a** Additional primary energy (TWh) in 2050 when the ban on gas boilers is implemented. **b** Additional installed capacity (GW) in 2050 when the ban on gas boilers is implemented. PV refers to photovoltaic. Source data are provided as a Source Data file.

Figure 3 shows the key uncertainties that undermine the climate objective in the absence of the ban. We show that the implementation of the ban significantly reduces reliance on biogas potential. Given that meeting residential gas demand is constrained by available biogas potential, the shift to heat pumps is driven by the ban on hedges against biogas supply shortages. This effect is exacerbated by the more efficient use of gas resources detailed in the precedent section. Furthermore, the regulatory nature of the ban ensures that the adoption of heat pumps is less dependent on uncertain demand-side factors, such as the ambition of subsidy policies or households' responsiveness. Conversely, without the ban, failure to meet climate targets may be prompted by an insufficient level of ambition in home insulation policies to reduce residual space heating demand, lower-than-expected household response to incentives (i.e., low price elasticity of heat pumps), or inadequate subsidies for low-carbon heating systems. Interactions among demand-side and supply-side uncertainties play a large role in the increased robustness of the ban, as evidenced by larger total-order indices compared to first-order indices. Overall, the ban appears as a more robust strategy to meet carbon neutrality against the uncertainty of various factors driving the decarbonization of the residential and energy sectors.

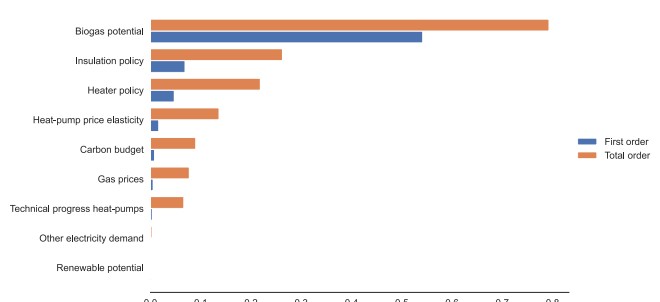

**Fig. 3 | Ranking of uncertainties undermining the achievement of climate targets in the current policy scenario compared to the ban.** First-order Sobol indices illustrate the share of variance explained by each uncertainty independently, while total-order Sobol indices represent the share of the variance explained by each uncertainty in interaction with other uncertainties. The latter can cumulatively exceed 1 (interaction terms are counted multiple times). Source data are provided as a Source Data file.

## A nuanced impact on total system costs

A comparison of total system costs is done across scenarios where both the ban and the current policy scenario achieve carbon neutrality. Total system costs are defined as the sum of annualized costs over the 2025–2050 period. All investment costs are annualized using a 3.2% discount rate, as recommended for public investment in France[24]. Supplementary Table 3 demonstrates that the choice of this parameter does not affect our findings.

Figure 4 shows that in the reference configuration, the scenario with the ban is slightly more expensive than the current policy scenario. Implementing the ban implies additional costs in heating systems as heat pumps, the most widely adopted system when the ban is implemented, are more expensive than gas boilers. In contrast, energy system investment and operation costs decrease. This cost decrease arises from the reduced primary energy need and optimized use of electricity capacities, as discussed above. Specifically, the ban relies on additional peaking power plant capacity while reducing the need for the more costly combination of renewable and battery storage capacities.

The comparison of total system costs across all uncertain scenarios, however, draws a different picture. In 49% of scenarios where both policy scenarios satisfy the carbon constraint, implementing the ban reduces total system costs. In particular, pessimistic assumptions on uncertain parameters require ambitious and expensive investments in energy system flexibility to accommodate the additional residual gas demand in the current policy scenario (Fig. 4b). The same factors that contribute to the increased robustness of the ban in achieving carbon neutrality, also make the scenario less costly (see Supplementary Fig. 5). This underlines that the current policy scenario only reduces total system costs compared to the ban scenario under specific conditions. Overall, in more than 75% of all scenarios - including those failing to meet carbon constraints and considered infinitely more costly -, implementing the ban results in lower total system costs. Our results highlight that relying solely on a reference configuration can be misleading, as it overlooks the nuanced cost-effectiveness of the ban amid existing uncertainties.

## Distributional impacts of the ban

We investigate the distributional consequences of implementing a ban in the reference configuration by comparing the cost incurred by different income groups and housing categories (occupancy status and

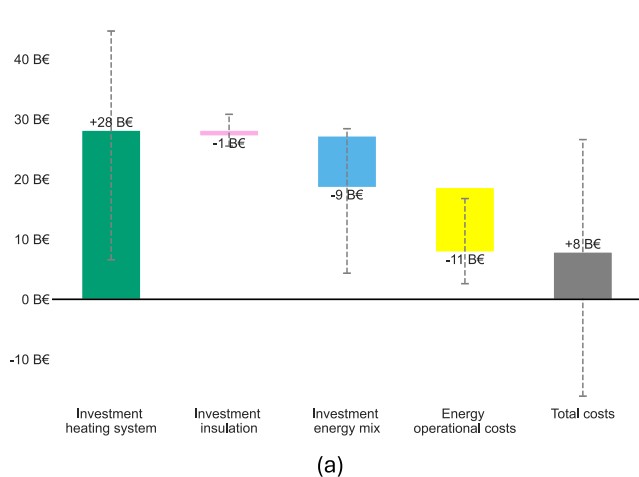

(a)

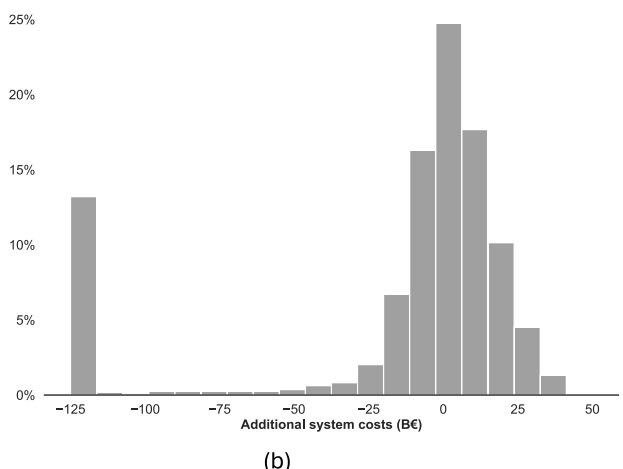

(b)

**Fig. 4 | Breakdown and distribution of additional cost when implementing the ban of gas boilers compared to the current policy scenario. a** Breakdown of additional cost in the Ban scenario, under the reference configuration. Error bars represent the 5th and 95th percentiles of the data set, including the 2566 scenarios that feature plausible energy systems. **b** Distribution of additional cost across

uncertainties. There are approximately 20% of the scenarios that incur significantly higher costs in the absence of the ban, including, for example, an exceptionally large amount of batteries. We winsorize at -125B€for readability. Source data are provided as a Source Data file.

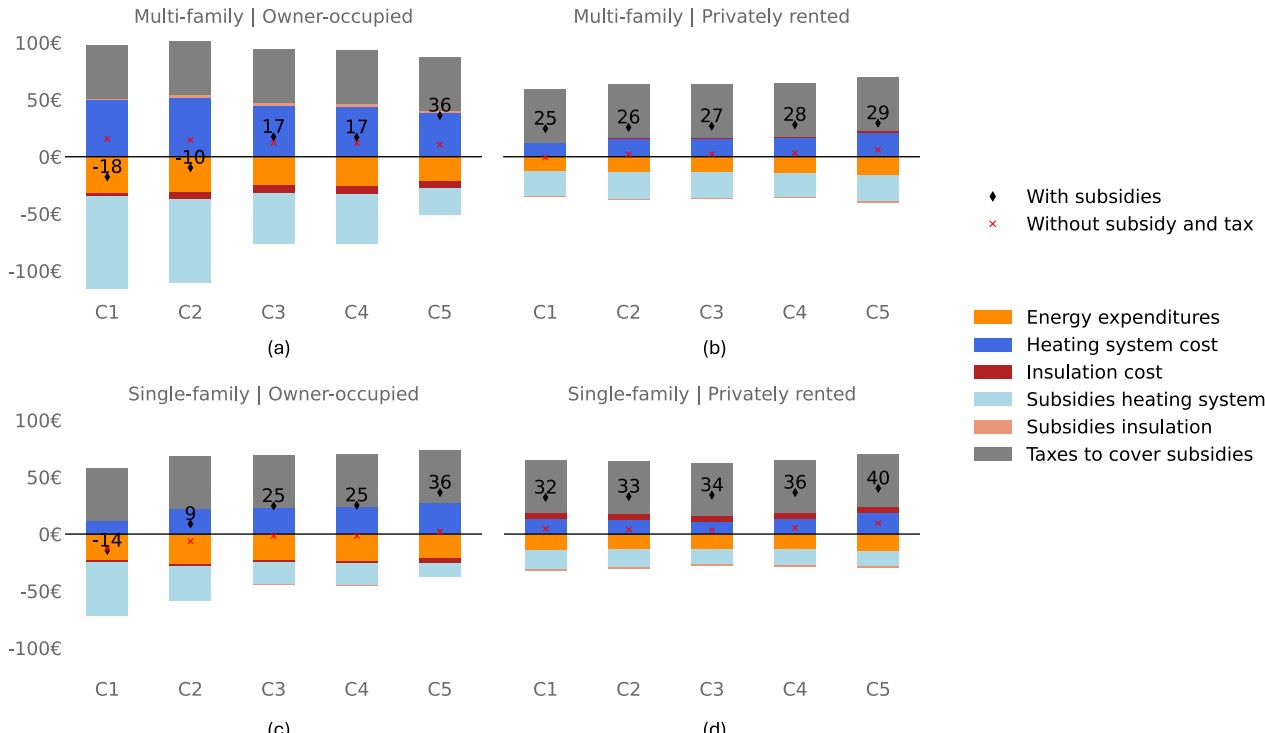

**Fig. 5 | Average yearly additional annual costs by household group under the reference configuration if the ban is implemented. a** Owner-occupier in multi-family dwellings. **b** Privately rented in multi-family dwellings. **c** Owner-occupier in single-family dwellings. **d** Privately rented in single-family dwellings. `C1' refers to the first income quintile, i.e., very low income, and `C5' refers to the last income quintile, i.e., very high income. A negative value means that the ban reduces household expenditure, while a positive value means that the ban increases household expenditure. Total cost is shown net of subsidies and taxes (black diamond) and without including these factors (red cross), in order to measure the strict effect of the ban before redistribution. Source data are provided as a Source Data file.

housing type) under the ban versus the current policy scenario. By doing so, we assess the marginal impact of the ban on households. Costs include heating system purchase costs and energy expenditure, supplemented by taxes meant to cover additional subsidy costs. We assume these taxes are evenly distributed among French households in a lump-sum manner, which is a standard approach in economic models[5,25,26]. Although these additional costs account for a small percentage of overall household energy costs, our analysis reveals significant disparities in the impact of the ban on households, with additional annual costs varying from -€18 to €40 across groups (see Fig. 5). These disparities are shaped by the financial impact of replacing the gas boiler on the intensive margin and the proportion of households affected by the ban on the extensive margin.

First, the financial impact of the ban depends on the profitability of adopting an alternative heating system, which varies widely among households. This variation primarily stems from differences in heating system choices and eligibility for subsidies. In short, adopting heat pumps is the only profitable choice, provided that subsidies are available to offset the purchase costs. Without substantial subsidies, or if households opt for wood fuel boilers or direct electric heating, the switch is not financially profitable for households. For owner-occupied households, the progressive nature of the French subsidy system, which adjusts the subsidy level to income, creates positive redistributive effects for low-income households (the first two income quintiles), while high-income households (the last two income quintiles) face adverse outcomes. Importantly, market and behavioral failures such as credit constraints and a strong present bias are prevalent among low-income households, leading them to choose less profitable investments such as direct electric systems. The subsidy design is, therefore, also instrumental in encouraging low-income households to invest in heat pumps, their most profitable option. In

contrast, for privately rented homes, investment decisions are made by landlords, who typically have higher incomes (see Supplementary Fig. 7) and are eligible for smaller subsidies. As a result, tenants who bear the investment cost of heating systems through increased rent, do not benefit from the subsidies that correspond to their level of income. This affects disproportionately low-income tenants (the first two income quintiles), as shown in Supplementary Fig. 6 through the relative impact on households' budgets. Consequently, while the implementation of the ban in France leads to progressive financial outcomes for owner-occupiers, it adversely impacts tenants. We also observe significant differences between housing types. Households in single-family homes, typically with more space, benefit more from the energy savings of switching to heat pumps, enhancing the profitability of their investment compared to those in multi-family homes. Conversely, some households in single-family homes may opt for wood boilers despite lower profitability. Overall, these mixed effects lead to a smaller range of distribution effects in single-family homes compared to multi-family homes.

Second, the impact of the ban, measured by the number of households needing to change their boilers, varies significantly across different groups. The differences are primarily across housing types rather than income levels. While the ban triggers additional government subsidies, we assume that these extra costs are financed by a lump-sum tax across all households. Consequently, households not directly impacted by the ban contribute to this tax, funding the subsidies without benefiting from them. Notably, in the current policy scenario, the share of gas boilers in privately rented and single-family homes is lower than in other groups (see Supplementary Fig. 11), implying that a smaller fraction of these households is affected by the ban and thus uses subsidies, despite bearing the cost of the lump-sum tax. This situation is particularly pronounced for low-income

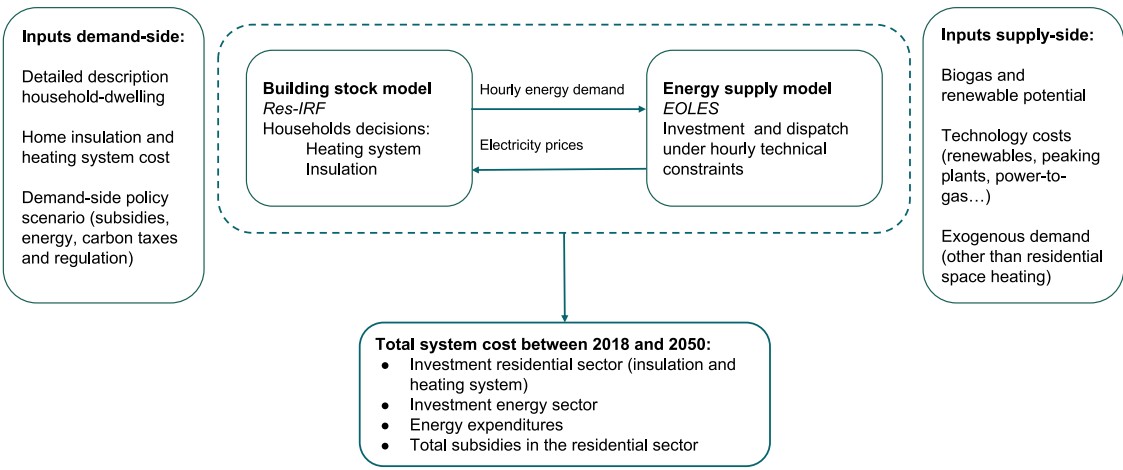

**Fig. 6 | Integrated modeling framework.** The framework integrates two detailed bottom-up models: (i) Res-IRF, which simulates energy demand for space heating, and (ii) EOLES, a comprehensive energy system model.

households in privately rented dwellings, who bear the tax burden without reaping the subsidy benefits aligned with their income level.

## Discussion

In this study, we present an evaluation of the highly debated ban on new fossil fuel boilers by assessing its robustness in achieving carbon neutrality under uncertainty, its cost-efficiency, and its distributional effects. First, the ban shifts the strategy for gas resource allocation from gas boilers to a combination of peaking power plants and heat pumps. This new allocation leads to an energy system that both reduces the need for primary energy generation and optimizes the utilization of electricity capacities. Second, we demonstrate that achieving carbon neutrality in the residential sector is highly uncertain under the current policy regime. In contrast, we show that the ban is a more robust strategy for achieving climate neutrality, showing no adverse effect on the electricity system while hedging against the lower-than-expected biogas potential. Third, despite costly investments in heating systems, the ban leads to lower total system costs over a large range of plausible futures. Fourth, we show that the distributional impacts are highly sensitive to the subsidy design for heat pumps and need to account for both income and occupation status. When coupled with the French existing subsidy framework, it mitigates vertical inequalities among owner-occupied households but does not extend to those in privately rented homes.

From a modeling perspective, we address a gap in the existing literature, which typically relies on simplified policy, such as shadow carbon pricing, and thus offers limited insights into climate policy design[27]. Specifically, we complement recent simulation studies that assess real-world policies in the residential sector[28–30], by also considering how these policies interact with the energy system. Our open-source modeling framework paves the way for investigating the impact of banning fossil fuel boilers in other economies like Germany or the Netherlands, which have the largest share of fossil fuel boilers among EU countries[9].

Choosing appropriate policy instruments for the transition to low-carbon heating systems is inherently difficult because of competing evaluation criteria[23]. We show that the ban on gas boilers is justified when moving beyond mere cost-effectiveness to consider the robustness of policies under uncertainty. Focusing only on a reference configuration can be misleading as it overlooks the nuanced cost impact of the ban amid existing uncertainties. This measure also involves trade-offs with distributional impacts, which can be mitigated through further research on the design of subsidies. Finally, our approach focuses on physical costs rather than the welfare criteria

often used in economics. Assessing the welfare impact of a ban in contexts with behavioral biases would, however, require more sophisticated models than those commonly used[31], at the expense of technical details.

## Methods
### Integrated energy demand-supply framework

Our framework integrates two detailed bottom-up models[17]: (i) Res-IRF, which simulates energy demand for space heating, and (ii) EOLES, a comprehensive energy system model (Fig. 6). Within a given time step, the exogenous policy scenario determines the final energy demand for residential space heating in the Res-IRF model. The EOLES model is subsequently run to optimize capacity investment and dispatch in the energy sector while meeting total energy demand and carbon budget. This process is then iterated in 5-year time steps, from 2020 to 2050. For a given period, wholesale electricity prices are endogenously computed as the levelized cost to meet demand from the previous period. The resulting prices are topped with exogenous energy taxes. The prices of other fuels (gas, oil, wood) are exogenous. Overall, the framework represents a high level of technological granularity both for the energy system (offshore, onshore, solar photovoltaic, nuclear, peaking plants, etc...) and residential sector (gas, oil, and wood boilers, direct electric and heat pumps).

Res-IRF is a dynamic microsimulation model of the energy demand for space heating in the French building stock[18]. The model was developed with the aim of improving behavioral realism. The model provides a comprehensive description of insulation levels (for walls, roofs, floors, and windows) and heating systems (heat pumps, electric heating, gas, oil and wood boilers). It simulates the evolution of energy consumption through three endogenous processes – the construction and demolition of buildings, the renovation of existing dwellings through insulation and fuel switching, and adjustments in heating behavior. Investments in energy efficiency are made by households and are influenced by the main economic costs and benefits, namely investment and financing costs, savings on energy bills, and subsidy amounts. In making these investments, households face various investment frictions, such as credit constraints, the inability of landlords to pass on energy efficiency investments to rents, decision frictions in collective housing, and hidden costs (e.g., the inconvenience of insulation work). The model also takes into account a gap between predicted and realized energy consumption to capture the much-discussed energy performance gap[32]. This wedge varies endogenously depending on energy efficiency improvements, energy prices, and household income and captures the rebound effect in

particular. The study presented here uses version 4.0 of the model. The data sources are listed in the Supplementary Information.

The model uses an hourly profile of heat pump efficiency to account for reduced performance during cold weather, which is crucial for determining peak demand. This efficiency is calculated based on the temperature difference between indoors and outdoors[33] and by assuming an indoor temperature of 55 Celsius degree (°C). By doing so, we capture the relationship between heat pump efficiency, space heating demand, and renewable energy generation.

The EOLES model is designed to optimize investment and operational decisions in France's energy system to satisfy a specified energy demand[34]. Its total costs cover annualized capital expenditures, maintenance expenses, and operational costs. The model is built on a comprehensive representation of various energy technologies. Electricity generation options include solar photovoltaic (PV), onshore and offshore wind, hydropower, open-cycle (OCGT) and combined-cycle gas turbines (CCGT), and nuclear power. Hydrogen production is achieved through water electrolysis. Gas sources range from fossil gas to biogas (produced via methanization or pyrogazeification) and synthetic methane through methanation. Energy storage is available in batteries, pumped-hydro storage, hydrogen storage in salt caverns, and methane storage in gas reservoirs. Technology dispatch operates on an hourly basis, accounting for weather-related fluctuations in supply and demand as well as flexibility requirements. Given the residential sector's significant reliance on gas, the gas-electricity interaction becomes essential. While the Res-IRF model focuses solely on residential energy demand, EOLES encompasses electricity demand across all end-use sectors. As such, non-residential energy demand projections (covering sectors such as commercial buildings, industry, transport, and agriculture) are integrated as exogenous inputs, drawn from the latest French Transmission System Operator projections[35]. This exogenous demand includes cooling requirements, making it unaffected by endogenous rebound effects. The analysis is confined to France, without accounting for cross-border energy exchanges. Further model specifics are detailed in Supplementary Information.

## Policy assessment

Our assessment is anchored within the carbon budget detailed in SNBC (Low Carbon National Strategy), France's national plan aiming for net zero emissions by 2050. Specifically, the allocated carbon budget for the residential sector, together with the power sector, is projected to be 26.5 Megaton of $CO_2$ ($MtCO_2$) annually by 2030, 20.5 $MtCO_2$ by 2035, 14.5 $MtCO_2$ by 2040, 9 $MtCO_2$ by 2045, and 4 $MtCO_2$ by 2050.

Our analytical framework is based on the comparison of scenarios that include the ban on gas boilers with counterfactual scenarios without the ban. Building on Vivier and Giraudet[36], we outline counterfactual scenarios that closely mimic the current policy mix for low-carbon heating in France. The current policy mix includes various energy efficiency measures, in particular a direct subsidy for heat pumps and wood fuel boilers of €4000 for low-income households (the first two income quintiles) and €2500 for high-income households (the last two income quintiles). It also includes mandatory insulation for private landlords, a carbon tax, and an oil boiler ban. The ban on gas boilers was introduced in 2025 and applied indiscriminately to single and multi-family dwellings. Concretely, when their heating system reaches the end of its lifetime, homeowners pick one replacement option among non-fossil fuel options, such as wood-fuel boilers, direct-electric, and heat pumps. District heating projections are determined exogenously, as they rely not on individual homeowner investments but on broader infrastructural investment decisions. We assume that homeowners only consider replacing their heating system when it is no longer working and, therefore, do not consider premature replacement. We also assume that the lifetime of heating systems remains constant over time, which means that we do not take into account repairs to extend the lifetime of a system. This effect could

reasonably be triggered by the implementation of the ban delaying the replacement of gas boilers.

Our analysis focuses on three key outcomes: the ability of a scenario to satisfy the carbon constraint, and, provided this constraint is met, the total system costs and a measure of distributional effects. Overall total system costs are defined as the sum of annualized costs over the 2025–2050 period. Building on Hirth et al.[37]'s work with the EMMA model, we use a 0% rate of pure time preference to give equal weight to all years when adding up annualized costs over the whole time horizon. The annualized system costs comprise both the investment and operational costs of the energy supply system, along with the costs associated with heating and insulation investments. The distributional indicator is defined as the average additional cost (or benefit) paid by the household group due to the introduction of a ban on gas boilers. These costs include the additional costs of the heating system net of subsidies, the energy costs, and a lump-sum tax meant to cover additional subsidy costs. We differentiate the costs according to income, occupation status (owner-occupied and private), and housing type (single-family and multi-family dwellings).

## Uncertainty assessment

The model processes rely on a large set of parameters, many of which are deeply uncertain. Such key uncertainties impact the supply energy system, the residential sector, and the other sectors (here only represented by the total electricity demand). Regarding the energy supply system, this corresponds to the potential for renewable technologies and renewable gas, as well as fuel prices. In the residential sector, it encompasses technological parameters such as the evolution of the efficiency and the price of heat pumps and behavioral parameters such as the average heat pump price elasticity. Table 1 summarizes the uncertain parameters and values used in this study. We perform extensive simulations over all possible combinations of uncertain parameters to estimate the distribution of outcomes.

We perform a global sensitivity analysis to identify the most influential vulnerabilities in the current policy scenario that are mitigated with the ban in place. We rely on variance decomposition methodology and we estimate Sobol indices based on our set of scenarios obtained by testing all combinations of uncertainty[38]. The variance decomposition is done to identify the uncertain determinants that increase the vulnerability of the current policy scenario.

For each scenario, we set variable $Y$ to the value 1 if the Ban scenario achieves carbon neutrality while the current policy scenario does not, −1 if the contrary holds, and 0 if both scenarios either meet or do not meet the carbon constraint. In our case, we actually never observe the −1 case. This outcome, therefore, directly measures the scenarios responsible for the increased vulnerability of the current policy policy scenario compared to the ban policy scenario. Since Sobol analysis is a variance decomposition method, the most influential drivers are, therefore, the parameters responsible for this increased vulnerability.

The first-order Sobol index $S_i$ is defined as equation (1).

$$S_i = \frac{\text{Var}\left(\mathbb{E}\left[Y|X_i\right]\right)}{\text{Var}(Y)} \tag{1}$$

Var corresponds to the variance, while $\mathbb{E}\left[Y|X_i\right]$ corresponds to the expectation of variable $Y$ conditioned on variable $X_i$. $X_i$ is a variable that corresponds to input variable $i$. $S_i$ measures the effect of varying $X_i$ alone on $Y$, but averaged over variations in other input parameters. A high $S_i$ value indicates that $X_i$ significantly influences the outcome by itself.

The total effect Sobol index $S_{T_i}$ is defined as equation (2).

$$S_{T_i} = 1 - \frac{\text{Var}(\mathbb{E}[Y|X_{-i}])}{\text{Var}(Y)} \tag{2}$$

$\mathbb{E}[Y|\boldsymbol{X}_{-i}]$ corresponds to the expectation of variable $Y$ conditioned on variable $\boldsymbol{X}_{-i}$, which corresponds to all input variables except for input variable $i$.

It measures the contribution to the output variance of $X_i$, including all variance caused by its interactions, of any order, with any other input variables. A low $S_{T_i}$ suggests that $X_i$ has minimal overall impact. Therefore, if $S_i$ is low but $S_{T_i}$ is high, it suggests that $X_i$ primarily affects the outcome through its interactions with other variables.

Other global sensitivity analyses include regression-based analysis[21]. These approaches typically assume linearity, attributing the residual sum-of-squares to variance unexplained by the model, due to nonlinear interactions. Given the significant nonlinear dynamics observed among uncertain drivers in our analysis, we opted for a variance decomposition methodology.

## Limitation

Here, we draw attention to four key limitations of our modeling approach.

First, our framework does not fully account for some costs associated with banning fossil fuel boilers. These include potential investments needed to expand the distribution network to enable increased heat pump uptake or the financial impact of stranded gas networks due to falling household demand for gas. We argue that these additional costs can be partially captured with high heat pump cost scenarios. Moreover, previous research has shown that residential electrification is expected to require far fewer distribution capacity additions than electric vehicle adoption Elmallah et al.[39]. We thus believe that this would not significantly alter our results.

Second, the building models overlook certain behavioral options. Following a ban on gas boilers, agents might choose to forego heating systems altogether or delay replacing their existing systems. Similar behavior has been observed in the automotive sector, where delayed vehicle replacement led to a rebound effect of 11% in energy savings[40].

Third, our analysis addresses the question of what would happen in France if we assess a ban on gas boilers. We take a positive approach, focusing on the outcomes rather than determining if the ban is superior to all other possible policy mixes. Further research could expand our analysis to compare different policy mixes with the implementation of the ban. In addition, we focus on one specific design of the ban-starting in 2030 and targeting all dwellings while other potential bans could, for example, target only standalone gas boilers.

Fourth, regulatory instruments, and bans in particular, can generate significant hidden costs, as they may conflict with consumers' preferences that are unobserved by the regulator. These hidden costs can be additional monetary costs, such as the laying of pipes or circuits, or non-monetary costs, such as the inconvenience of finding out about a new heating system, the cost of obtaining information, or the inconvenience during the works[41]. We do not include these hidden costs in our cost analysis primarily because they are difficult to identify without further empirical research. Moreover, these costs could fluctuate over time with changes in consumer preferences and may also be directly affected by the implementation of the ban. However, they would amount to additional costs for heat pumps and can again be partially captured by the high-cost scenario for heat pumps. Such potential additional costs, though they could reduce the cost-effectiveness of banning gas boilers, would, however, not alter the conclusion that the ban is critical to meet climate targets. Overall, further research could move away from the 'accounting approach' used here to assess cost-effectiveness towards a 'welfare approach' that takes into account the unobserved utility (i.e., including hidden cost) of households in adopting a particular technology[26].

## Reporting summary

Further information on research design is available in the Nature Portfolio Reporting Summary linked to this article.

## Data availability

The results data generated in this study have been deposited in the Zenodo repository under accession code https://doi.org/10.5281/zenodo.14039683. Source data are provided in this paper.

## Code availability

The code of the integrated modeling framework has been deposited in the Zenodo repository under the accession code https://doi.org/10.5281/zenodo.14039620.

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

## Acknowledgements
This research was supported by the Agence Nationale de la Recherche (ANR) under project PREMOCLASSE (grant ANR-19-CE22-0013-01), awarded to L.V., as well as by ADEME and the Chaire Stress Test, awarded to C.E. Funding and support were also provided by the Palladio Foundation, awarded to L.V. We thank Adrien Fabre, Louis-Gaëtan Giraudet, Simon Jean, Philippe Quirion, Thibault Briera, Romain Fillon, Céline Guivarch for useful comments and suggestions on previous versions of this paper.

## Author contributions
C.E. and L.V. conceptualized, designed the study, curated data, supported model development, ran the model, created visualizations, wrote the original draft, and reviewed and edited the manuscript.

## Competing interests
The authors declare no competing interests.
