## [Peer Review File · Nature Communications]

REVIEWER COMMENTS

Reviewer #1 (Remarks to the Author):

The manuscript “Banning new gas boilers as a hedge against the limited availability of renewable gas supply” uses a modelling approach to study the impacts of an introduction of a ban on fossil boilers in France. It analyses impacts on greenhouse gas emissions, cost effectiveness and distributional effects.

Key results

One of the key results of the paper is that it is more likely that carbon neutrality is achieved with a ban on installing fossil boilers as compared to a situation without such a ban. This is substantiated by 11 664 scenario calculations and comparing the results of the scenarios with and without the ban.

The paper further analyses system costs of different scenarios that meet carbon neutrality and finds that 1) costs are higher with a ban as compared to the reference; 2) In 49% of scenarios where both policy scenarios meet carbon neutrality, implementing the ban reduces total system costs

Finally, the paper analyses distributional effects when switching from gas boilers to heat pumps. In this last part of the analysis is not fully clear how it is linked to the various scenarios calculated in the first part of the paper.

Significance of the results

The result that it is more likely to reach carbon neutrality with a ban of fossil boilers than without is somewhat unsurprising: The authors assume that with the ban no new heating systems using fossil fuels are installed. It is rather obvious that in such a case it is much more likely to reach carbon neutrality than in any scenario in which fossil fuel boilers are still installed. For that reason, the significance of the finding is limited.

Regarding the analysis of cost-effectiveness, the results are somewhat ambiguous due to the large number of different scenarios. While it is good that the authors account for uncertainties, the wide range of possible results leads to findings that have limited significance as no clear conclusion can be drawn regarding whether a ban is cost-effective or not. Another key limitation is that the costs for expanding the electricity system are not included.

Regarding the analysis of distributional effects, the results strongly depend on the design of the subsidy schemes, which provide higher subsidies to low-income households. While the authors do acknowledge that the results depend on the subsidy scheme, the framing of the results is that the distributional effects of the ban are analysed. This is somewhat misleading, as the main results are more strongly influenced by the subsidies than by the ban. The authors do not provide any insights of distributional effects of the ban as such.

Consistency of claims and conclusions

The authors claim to “present the first evaluation of the highly debated ban on new fossil fuel boilers by assessing its robustness in achieving carbon neutrality under uncertainty, its cost-efficiency, and its distributional effects.” There are some limitations in this context:

- 1) Due to the large number of scenarios and the very broad range of uncertainties, the results regarding the robustness in achieving carbon neutrality provide limited benefit, as the fact that it is more likely to reach climate neutrality with a ban is rather obvious (see explanations above)
- 2) Regarding cost-effectiveness, the results remain rather vague due to the large number of scenarios with differing results. This means no clear conclusions can be drawn.
- 3) Regarding distributional effects, as pointed out above, the results strongly depend on the subsidies. The authors do not present how the other scenarios (the ones without a ban) perform with respect to distributional effects, however it is very likely that the distributional effects would be very similar (as they mainly depend on the subsidies). The claim “we show that the implementation of the ban, when coupled with the French existing subsidy framework, mitigates vertical inequalities among owner-occupied households but does not extend to those in privately rented homes.” is true but would need to be seen in context with the policy alternatives (for which my guess would be that the same holds). I would recommend to separate the analysis of distributional effects and treat it in a separate paper, as it is only marginally related to the boiler ban.

The authors dedicate one section to the efficiency of the energy system. The claim stated in the title of the section (“Widespread heat pump adoption leads to a more efficient energy system”) is not sufficiently substantiated in the text. The references to the supplementary material do not explain sufficiently how the authors define system efficiency and how the heat pump adoption contributes to increasing efficiency.

Validity and methodological soundness

The work has the limitations pointed out above. These are not necessarily flaws and they do not prohibit publication as such, however they do limit the importance of the results for the field and thus the likelihood that the paper will significantly move forward the understanding of policy impacts around banning boilers. It may therefore be worth to consider if there are other more specialized journals that are better suited for the content of the work.

The authors use a sound modelling approach to explore numerous pathways for the buildings sector. They acknowledge the uncertainties associated with the assumption by creating a broad set of scenarios covering numerous combinations of different parameters. However, key parameters relevant for the system costs are not varied and no sensitivities are presented. For example, Figure 8 suggests that for heat pumps an efficiency of 2,5 has been assumed, which seems rather on the low side (or should be substantiated with empirical data on the average efficiency of the French heat pump stock). Other key parameters are the purchase prices presented in Table 10. It would be helpful to provide more detail on empirical data substantiating these parameters and on the uncertainties that arise from their variations. Another key parameter for the calculation of cost-effectiveness is the discount rate, where the authors state that a value of 3.9% is used. This is also a key parameter, and it should be substantiated why this value has been chosen, and how higher and lower discount rates would influence the results.

Generally, models that aim at mimicking investment behaviour of individuals have important limitations, as the empirical basis to support the parametrization of the models is weak, compared to the complexity of the problem (millions of households in a very heterogeneous building stock and varying policy framework). It is therefore questionable if such models accurately reflect the reaction of individuals to policy changes. However, the use of such models reflects the standards in the field, such that this is not per se a limitation for the article under consideration.

Some of the data used in the model seem somewhat outdated, e.g. the cost data presented in Table 10 is referenced to the year 2020 and seems to underestimate actual costs.

For financing the subsidies, the authors assume a lump-sum tax across all households. This assumption would need more explanation. Why are the subsidies not financed through the state budget? In the lump-sum case low-income households seem to be much more affected.

Some claims are not well substantiated with data and may strongly influence the results. For example, the authors state the following: "Importantly, credit constraints and a strong present bias are prevalent among low-income households, leading them to choose less profitable investments such as direct electric systems. -> it does not seem that any reliable empirical data exists that would cover the situation described in the paper (replacement of boiler by heat pump)

Level of detail provided by the authors

The authors provide details on the modelling framework and refer to other papers where more details on the models are provided.

It is not clear, which scenarios (or which system costs) are used for the analysis of distributional effects. As stated in the paper, system costs vary considerably. The authors would need to explain in more details which of the scenarios are used for the analysis and how the results would change when using different scenarios.

Other remarks

The title rather prominently highlights the “availability of renewable gas supply”, which in the paper is only one of the many parameters that are varied in the scenarios and the availability of renewable gas is not considered in the research questions outlined in the introduction. I would recommend to either change the title (preferred option) or provide more discussion on the availability of renewable gas supply and how the ban affects this. The same holds for the abstract, where renewable gas supply features in two sentences (“Taking France as a case study, we find that heat pump adoption shifts gas use from heating demand to electricity generation, which is a more efficient use of low-carbon biogas from a wholesystem perspective. Heat pump adoption therefore provides a hedge against short supply of low-carbon gas.”), either provide a separate section discussing this topic or remove/shorten in abstract (preferred option).

The following sentence is outdated should be adapted and refer to the final version of the EPBD adopted in 2024: “Furthermore, a recent agreement in the Energy Performance Building Directives mandates that Member States implement measures to completely phase out fossil fuel heating and cooling by 2040 (Commission, 2023)”.

Reviewer #2 (Remarks to the Author):

The paper reveals that banning new gas boilers significantly boosts the adoption of heat pumps, leading to a shift from gas to electricity for heating, which enhances system-wide efficiency. The work provides critical insights into the effectiveness of regulatory measures for promoting low carbon heating technologies and achieving carbon neutrality, offering valuable guidance for energy policy and residential sector decarbonization.

The findings align with existing literature on the benefits of heat pumps and the challenges of residential sector decarbonization, but this study advances the discussion by providing more nuanced policy insights.

There are some limitations, such as assumptions about replacement timing, behavioural responses, and hidden costs. The recognition of these limitations within the work enhances its credibility and transparency.

The methodology is robust and meets expected standards. Scenario analysis aims to capture key uncertainties and interactions between the residential and energy sectors. However, the placement of the methods section at the end of the paper is unconventional and may lead to confusion. It would be more effective to present the methods earlier, providing a clear framework before discussing the results and implications.

Additionally, the work supports its conclusions and claims well throughout the paper whereas including a dedicated conclusion section would help to succinctly draw and highlight the key conclusions of the study.

Reviewer #2 (Remarks on code availability):

The repository includes detailed instructions for installation, setup, and running simulations, ensuring reproducibility and ease of use. The methodology involves creating scenarios and running scripts to analyze policy impacts on energy systems. I couldn't manage to install and run the code due to technical problems occurred in personal PC but the repository appears to be functional and well-documented.

Reviewer #1 (Remarks to the Author)

Significance of the results

We greatly appreciate the valuable feedback from Reviewer 1 and have revised our manuscript to enhance the presentation of our findings on banning fossil fuel boilers. We believe that the revised text now clearly emphasizes how our study significantly advances the current understanding of this topic. Below we provide a point-by-point response to Reviewer 1's comment.

The result that it is more likely to reach carbon neutrality with a ban of fossil boilers than without is somewhat unsurprising: The authors assume that with the ban no new heating systems using fossil fuels are installed. It is rather obvious that in such a case it is much more likely to reach carbon neutrality than in any scenario in which fossil fuel boilers are still installed. For that reason, the significance of the finding is limited.

We acknowledge that we did not sufficiently emphasize the significance of our findings in the original draft, which may have made the results appear overly intuitive. However, we believe that there is no scientific consensus on this issue and that only careful numerical application can bring meaningful insight to this important question.

In particular, there is ongoing debate about the capacity of the electric system to rapidly accommodate the additional electricity demand resulting from a ban on fossil fuel boilers. Theoretically, if the electricity system were unable to absorb this demand due to overly stringent and swift measures, this could prompt an increased reliance on fossil fuels, which would undermine the goal of achieving carbon neutrality. Moreover, it is conceivable that a ban on new gas boilers may not be necessary to meet climate targets in the energy-residential sector if current incentives in the residential sector were adequately designed.

Main text modifications: We better introduce this debate in our introduction to ensure that the reader can understand the trade-offs involved : *“A major concern of the ban on gas boilers is the induced energy system externalities. Specifically, a rapid increase in space heating electricity demand concentrated during peak load, due to stringent measures, could require further investments in the electricity sector, increasing overall costs and hampering the ability to achieve carbon neutrality.”*

Our calculation demonstrates (i) that even with the current ambitious policy package, climate targets are not fully met; and ii) that the electricity system can absorb this additional demand without adverse effects. These findings allow us to conclude that the ban does increase the probability of achieving climate targets.

Additionally, our framework, which accounts for multiple demand-side and supply-side uncertainties, enables us to identify in a robust and quantitative manner a key uncertainty that

underpins the superiority of the ban in meeting ambitious climate targets: the potential of biogas. This finding is uniquely possible due to our detailed modeling approach, which explicitly incorporates uncertainty—a crucial aspect of our work.

Main text modifications: We better emphasize those new findings in our conclusion: “ *Second, we demonstrate that achieving carbon neutrality in the residential sector is highly uncertain under the current policy regime. In contrast, we show that the ban is a more robust strategy for achieving climate neutrality, showing no adverse effect on the electricity system while hedging against the lower-than-expected biogas potential.*”

Regarding the analysis of cost-effectiveness, the results are somewhat ambiguous due to the large number of different scenarios. While it is good that the authors account for uncertainties, the wide range of possible results leads to findings that have limited significance as no clear conclusion can be drawn regarding whether a ban is cost-effective or not. Another key limitation is that the costs for expanding the electricity system are not included.

We consider the nuanced impact of the ban on total system costs to be a significant result in its own right. This issue has not yet been fully addressed in the literature, although it is often argued that regulatory instruments are less cost-effective than incentives. Our study emphasizes that the impact on costs is actually not straightforward. Accurately estimating these costs requires an integrated approach, as the ban has significant implications for the whole energy system.

Our results show that in 75% of scenarios - accounting also for scenarios where the current policy scenario does not achieve carbon neutrality - costs are lower with the ban. This demonstrates that relying solely on a reference configuration may be misleading, as it overlooks the nuanced cost-effectiveness of the ban amid existing uncertainties.

Main text modifications: We have refined the language in this section to be more convincing about the added value of examining a broad range of uncertainties to explore cost implications: “*Overall, in more than 75% of all scenarios - including those failing to meet carbon constraints and considered infinitely more costly, implementing the ban results in lower total system costs. Our results highlight that relying solely on a reference configuration can be misleading, as it overlooks the nuanced cost-effectiveness of the ban amid existing uncertainties.*”

In acknowledging the limitation of not including transmission and distribution costs in our analysis, we recognize that a more comprehensive study should aim to incorporate these factors. However, previous research indicates that residential electrification is likely to require significantly fewer distribution capacity additions compared to electric vehicle adoption (Elmallah et al., 2022). Therefore, we consider this assumption reasonable and unlikely to affect our main results.

Main text modifications: We add the following in the text in the Limitation subheading of the Method section: “*Moreover, previous research has shown that residential electrification is*

expected to require far fewer distribution capacity additions than electric vehicle adoption (Elmallah et al., 2022). We thus believe that this would not significantly alter our results.”

Regarding the analysis of distributional effects, the results strongly depend on the design of the subsidy schemes, which provide higher subsidies to low-income households. While the authors do acknowledge that the results depend on the subsidy scheme, the framing of the results is that the distributional effects of the ban are analysed. This is somewhat misleading, as the main results are more strongly influenced by the subsidies than by the ban. The authors do not provide any insights of distributional effects of the ban as such.

We thank Reviewer 1 for their insightful comments on distributional impacts. We acknowledge that the previous version of the text did not clearly explain what this analysis represents and which scenarios were used. We have now clarified that this analysis is based on the Reference configuration and compares a scenario with the ban and a scenario without the ban. By doing so, our findings isolate the impact of the ban itself, showing that implementing a ban in conjunction with existing subsidies helps mitigate vertical inequalities among owner-occupied households but does not extend these benefits to those in privately rented homes.

Main text modifications: We have improved the introduction of our distributional analysis: *“We investigate the distributional consequences of implementing a ban in the reference configuration by comparing the cost incurred by different income groups and housing categories (occupancy status and housing type) under the ban versus the current policy scenario. By doing so, we assess the marginal impact on households of introducing the ban.”*

While our results are strongly influenced by the current subsidy design, we believe this is an important result, as most high-income and temperate countries are subsidizing heat pumps. Moreover, as scenarios without the ban and without subsidies cannot achieve carbon neutrality, we are unable to measure the marginal impact independent of subsidy effects.

Main text modifications: *“Fourth, we show that the distributional impacts are highly sensitive to the subsidy design for heat pumps, requiring consideration of both income and occupation status.”*

Consistency of claims and conclusions

The authors claim to “present the first evaluation of the highly debated ban on new fossil fuel boilers by assessing its robustness in achieving carbon neutrality under uncertainty, its cost-efficiency, and its distributional effects.” There are some limitations in this context:

1) Due to the large number of scenarios and the very broad range of uncertainties, the results regarding the robustness in achieving carbon neutrality provide limited benefit, as the fact that it is more likely to reach climate neutrality with a ban is rather obvious (see explanations above)

Please see our response to this point provided above.

2) Regarding cost-effectiveness, the results remain rather vague due to the large number of scenarios with differing results. This means no clear conclusions can be drawn.

Please see our response to this point provided above.

3) Regarding distributional effects, as pointed out above, the results strongly depend on the subsidies. The authors do not present how the other scenarios (the ones without a ban) perform with respect to distributional effects, however it is very likely that the distributional effects would be very similar (as they mainly depend on the subsidies).

As mentioned above, we did present the comparison between the scenario with and without the ban to isolate the effect.

The claim “we show that the implementation of the ban, when coupled with the French existing subsidy framework, mitigates vertical inequalities among owner-occupied households but does not extend to those in privately rented homes.” is true but would need to be seen in context with the policy alternatives (for which my guess would be that the same holds). I would recommend to separate the analysis of distributional effects and treat it in a separate paper, as it is only marginally related to the boiler ban.

We appreciate the reviewer’s suggestion to develop a separate paper on the distributional consequences, but we believe it is crucial to provide the Nature Communications audience with a comprehensive overview of the main implications of the ban. Distributional outcomes are a critical aspect when evaluating the implementation of a ban, and we believe our analysis contributes interestingly to this discussion. We agree that further research should focus on exploring how different subsidies could yield even better distributional outcomes as we discuss at the end of the paper.

We mention in the Discussion section that *“This measure also involves trade-offs with distributional impacts, which can be mitigated through further research on the design of subsidies.”*

The authors dedicate one section to the efficiency of the energy system. The claim stated in the title of the section (“Widespread heat pump adoption leads to a more efficient energy system”) is not sufficiently substantiated in the text. The references to the supplementary material do not explain sufficiently how the authors define system efficiency and how the heat pump adoption contributes to increasing efficiency.

We thank the reviewer for pointing out the lack of clarity regarding the impacts on the energy system. We have revised the first section of the **Results** by i) removing the concept of “efficient energy system” which was not precisely defined, and ii) discussing successively the two indicators that allow us to analyze changes in the energy system.

Main text modifications: The section is now written as follows: “*Banning gas boilers leads to significant transformations within the energy system by (i) reducing primary energy need, and (ii) improving the capacity factors.*”

First, Figure \ref{fig:generation} shows that the system relies on less primary energy to deliver the same energy services. By 2050, the ban reduces the primary energy requirement by 12 TWh. These shifts are driven by different strategies for allocating gas resources, which are constrained by carbon constraints and limited biogas potential. While low-carbon gas is currently used in gas boilers, it could be redirected to peaking power plants that support electric heating systems if the ban is enforced. Overall, we find that the combination of peaking power plants and heat pumps meets energy service demands more efficiently, as shown in Supplementary Figure \ref{fig:simplified_schema_interaction}.

Second, meeting peak demand with peaking power plants eliminates the need for the combination of renewable capacity and battery storage as a flexibility solution. Specifically, Figure \ref{fig:capacity} demonstrates that the ban avoids the installation of 12 GW of renewable capacity (offshore wind and solar PV) and 3 GW of battery storage, while instead requiring an additional 12 GW of peaking plants (Table \ref{table:summary-results}). This reduction in renewable capacity leads to a more efficient use of nuclear power as a base-load generator, thereby increasing its capacity factor.”

Additionally, we have modified the title of the section: “*A ban addresses energy service demands more efficiently*”

Finally, we have changed how we state our contributions at the end of the **Introduction** section: “*First, we demonstrate that the additional electricity demand resulting from the implementation of the ban does not have any adverse effects on the electricity system. Instead, it leads to reduced primary energy requirements and improved capacity factors for power plants.*”

Validity and methodological soundness

The work has the limitations pointed out above. These are not necessarily flaws and they do not prohibit publication as such, however they do limit the importance of the results for the field and thus the likelihood that the paper will significantly move forward the understanding of policy impacts around banning boilers. It may therefore be worth to consider if there are other more specialized journals that are better suited for the content of the work.

Space heating is responsible for 20% of greenhouse gas emissions in the European residential sector. Our paper is the first to specifically address the issue of banning gas boilers, a topic that is highly debated and likely to interest a wider audience than that of a more specialized journal. In addition, from a methodological point of view, we think our framework is at the frontier of the assessment of realistic policies, a recent expectation for energy-economy models.

The authors use a sound modelling approach to explore numerous pathways for the buildings sector. They acknowledge the uncertainties associated with the assumption by creating a broad set of scenarios covering numerous combinations of different parameters.

However, key parameters relevant for the system costs are not varied and no sensitivities are presented.

We thank Reviewer 1 for suggesting the inclusion of these additional key parameters. While we agree that incorporating a broad range of uncertainties is essential for robust policy assessment, it is also necessary to manage the complexity of the study. Adding another dimension would result in at least 11,664 additional scenarios, which would significantly increase computational time (with each scenario taking approximately 30 minutes in our recursive framework, this would require an estimated 12 days of computation, even with parallel processing). This is not an issue in itself, but it does illustrate why we have chosen to limit our study to the current number of dimensions, which initial exploration indicated as the most critical.

For example, Figure 8 suggests that for heat pumps an efficiency of 2.5 has been assumed, which seems rather on the low side (or should be substantiated with empirical data on the average efficiency of the French heat pump stock).

We also thank Reviewer 1 for highlighting the importance of heat pump efficiency. In pre-publication work, we tested the sensitivity of heat pump efficiency and found no significant impact on the marginal effects of the ban. Consequently, we decided not to include it in the current paper to reduce complexity. However, this feedback also indicates that we did not adequately describe our approach to estimating heat pump efficiency, which is not a constant value of 2.5 but varies with outdoor temperature across different days.

Main text modifications: We add this sentence in the Methods section to clarify this aspect: *“We use an hourly profile of heat pump efficiency to account for reduced performance during cold weather, which is crucial for determining peak demand. This efficiency is calculated based on the temperature difference between indoors and outdoors \autocite{staffell_review_2012} and by assuming an indoor temperature of 55°C. By doing so, we capture the relationship between heat pump efficiency, space heating demand, and renewable energy generation.”*

Additionally, we have included a sentence in Supplementary Figure 8 to indicate that the value of 2.5 is provided as an example. *“In the model, heat pump efficiency is not a constant value but varies with outdoor temperature across different days. For simplicity in this figure, we use a value of 2.5, which represents the lower end of the range but still illustrates the higher efficiency of the system.”*

Other key parameters are the purchase prices presented in Table 10. It would be helpful to provide more detail on empirical data substantiating these parameters and on the uncertainties that arise from their variations.

Regarding the purchase price of heating systems, we have used data from a reference study conducted in France (RTE and ADEME, 2020). We have now provided additional details in the Supplementary Information.

“Costs are consistent with the JRC DataSet \autocite{hofmeister techno-economic_2017} and a previous modeling study \autocite{knobloch_fttheat_2021}.”

We believe the critical factor in our analysis is the cost differential between gas boilers and heat pumps, rather than the absolute cost of each technology. This is why we chose to include the learning effect of heat pumps as a key uncertain parameter. This decision is supported by evidence that price variations are likely to result from reductions in heat pump costs due to technological progress (Wess et al., 2010). Therefore, we believe that our study adequately addresses uncertainties surrounding heating system costs.

Another key parameter for the calculation of cost-effectiveness is the discount rate, where the authors state that a value of 3.9% is used. This is also a key parameter, and it should be substantiated why this value has been chosen, and how higher and lower discount rates would influence the results.

Finally, we agree that the discount rate is an important parameter that was not fully addressed in our initial study. The 3.9% discount rate we used in the distributional impacts analysis represents the average interest rate for energy renovation in France, not the social discount rate. We have clarified this distinction. For our cost-effectiveness analysis, we use the reference social discount rate recommended by authorities of 3.2%. We have now included simulations using five different discount rates, as shown in the Supplementary Tables (Supp. Fig. 3). Due to the distinct nature of social discount rates which are more of a normative parameter than an uncertain one, we did not include them in the primary uncertainty analysis.

Main text modifications: *“All investment costs are annualized using a 3.2% discount rate, as recommended for public investment in France \autocite{ni_revision_2021}. Supplementary Table \ref{table:sensitivity-discount-rate} demonstrates that the choice of this parameter does not affect our findings.”*

Generally, models that aim at mimicking investment behaviour of individuals have important limitations, as the empirical basis to support the parametrization of the models is weak, compared to the complexity of the problem (millions of households in a very heterogeneous building stock and varying policy framework). It is therefore questionable if such models accurately reflect the reaction of individuals to policy changes. However, the use of such models reflects the standards in the field, such that this is not per se a limitation for the article under consideration.

Some of the data used in the model seem somewhat outdated, e.g. the cost data presented in Table 10 is referenced to the year 2020 and seems to underestimate actual costs.

We believe that the cost data we used are consistent with the reference literature on the subject (JRC DataSet). Specifically, heat pumps—which in our analysis simulate water-to-air technology since they are typically installed in dwellings with existing hydraulic systems for heat emission—cost between €8,000 and €16,000 in France in 2024, depending on the surface area and manufacturer¹. In Supplementary Table 10, we add “Costs are consistent with the JRC DataSet \autocite{hofmeister techno-economic_2017} and a previous modeling study \autocite{knobloch_fttheat_2021}.”.

For financing the subsidies, the authors assume a lump-sum tax across all households. This assumption would need more explanation. Why are the subsidies not financed through the state budget? In the lump-sum case low-income households seem to be much more affected.

We agree with the reviewer that more explanation is needed. We assume that subsidies are financed by the state, which necessitates raising taxes to cover these costs. Therefore, households face additional taxes designed to finance the subsidies. Specifically, we assume that these taxes are collected in a lump-sum manner, meaning each household faces the same amount of tax. As this is not the central focus of the paper, we chose not to explore other possible methods for financing the subsidies.

Main text modifications: We have clarified how subsidies are financed in the distributional analysis section: “We assume these taxes are evenly distributed among French households in a lump-sum manner, which is a standard approach in economic models”

Some claims are not well substantiated with data and may strongly influence the results. For example, the authors state the following: “Importantly, credit constraints and a strong present bias are prevalent among low-income households, leading them to choose less profitable investments such as direct electric systems. -> it does not seem that any reliable empirical data exists that would cover the situation described in the paper (replacement of boiler by heat pump)

We believe the reviewer may have been referring to “direct electric systems” rather than “heat pumps” in the last sentence of their comment. We acknowledge that we do not have direct empirical data demonstrating this situation. The objective of our model is to simulate complex scenarios, such as heating system decisions influenced by various frictions, which have not occurred in the past but are likely to occur in the future. In particular, we anticipate that households in France who cannot afford heat pumps may opt for direct electric systems instead. For example, Stolyarova (2016) shows that low-income households tend to undervalue energy savings compared to higher-income households, leading them to make suboptimal investment decisions from a rational perspective. We refer to Vivier and Giraudet (2024) for additional information on household investment decisions.

¹<https://particuliers.engie.fr/depannages-services/conseils-equipements-chauffage/conseils-pompe-a-chaleur/cout-pompe-a-chaleur.html>
<https://www.leroymerlin.fr/produits/chauffage-et-ventilation/pompe-a-chaleur/>

We add: *“Importantly, market and behavioral failures such as credit constraints and a strong present bias are prevalent among low-income households, leading them to choose less profitable investments such as direct electric systems.”*

It is also important to note that direct electric systems are more common in France compared to other European countries, where such a trend might appear unexpected. Furthermore, data indicates that lower-income households in France are more likely to reside in dwellings heated by direct electric systems. As a result, these households may choose to replace older electric systems with newer ones, continuing this trend.

Level of detail provided by the authors

The authors provide details on the modelling framework and refer to other papers where more details on the models are provided. It is not clear, which scenarios (or which system costs) are used for the analysis of distributional effects. As stated in the paper, system costs vary considerably. The authors would need to explain in more details which of the scenarios are used for the analysis and how the results would change when using different scenarios.

We thank the reviewer for this comment. Our distributional effects analysis is done under the reference configuration. The reasons are twofold: 1) as this analysis is already very rich, we believed adding an additional dimension would complicate the reader’s comprehension, and 2) although the magnitude of these distributional effects may vary across other scenarios, the overall effects remain consistent. We have modified the corresponding section accordingly to explicitly state the configuration we use in the analysis.

Main modifications in the text: *“We investigate the distributional consequences of implementing a ban in the reference configuration...”*

Other remarks

The title rather prominently highlights the “availability of renewable gas supply”, which in the paper is only one of the many parameters that are varied in the scenarios and the availability of renewable gas is not considered in the research questions outlined in the introduction. I would recommend to either change the title (preferred option) or provide more discussion on the availability of renewable gas supply and how the ban affects this.

The same holds for the abstract, where renewable gas supply features in two sentences (“Taking France as a case study, we find that heat pump adoption shifts gas use from heating demand to electricity generation, which is a more efficient use of low-carbon biogas from a wholesystem perspective. Heat pump adoption therefore provides a hedge against short supply of low-carbon gas.”), either provide a separate section discussing this topic or remove/shorten in abstract (preferred option).

We agree with the reviewer that both the title and the abstract of the paper were overly focused on the availability of renewable gas supply. We have now changed the second section to discuss more clearly how the ban leads to increased robustness in the face of a combination of demand-side and supply-side uncertainties. We also changed the title and the abstract accordingly. Specifically, our new title is: "Banning new gas boilers as a no-regret mitigation option"

The following sentence is outdated should be adapted and refer to the final version of the EPBD adopted in 2024: "Furthermore, a recent agreement in the Energy Performance Building Directives mandates that Member States implement measures to completely phase out fossil fuel heating and cooling by 2040 (Commission, 2023)".

We thank the reviewer for highlighting this outdated reference. In the revised draft, we have updated the citation to refer to the final version of the EPBD adopted in April 2024.

Reviewer #2 (Remarks to the Author):

The paper reveals that banning new gas boilers significantly boosts the adoption of heat pumps, leading to a shift from gas to electricity for heating, which enhances system-wide efficiency. The work provides critical insights into the effectiveness of regulatory measures for promoting low carbon heating technologies and achieving carbon neutrality, offering valuable guidance for energy policy and residential sector decarbonization.

The findings align with existing literature on the benefits of heat pumps and the challenges of residential sector decarbonization, but this study advances the discussion by providing more nuanced policy insights.

There are some limitations, such as assumptions about replacement timing, behavioural responses, and hidden costs. The recognition of these limitations within the work enhances its credibility and transparency.

The methodology is robust and meets expected standards. Scenario analysis aims to capture key uncertainties and interactions between the residential and energy sectors. However, the placement of the methods section at the end of the paper is unconventional and may lead to confusion. It would be more effective to present the methods earlier, providing a clear framework before discussing the results and implications.

We thank the reviewer for this comment. However, according to the guidelines of *Nature Communications*, the Methods section should be placed after the Results section. We understand that this arrangement might make it more challenging for some readers to follow our approach. To address this, we have included a short explanation of the main key points of our methodology at the end of the Introduction, prior to introducing our main results.

Additionally, the work supports its conclusions and claims well throughout the paper whereas including a dedicated conclusion section would help to succinctly draw and highlight the key conclusions of the study.

We strive to address these concerns in the first paragraph of the Discussion following the guidelines of *Nature Communications*. Specifically, we state our key contributions:

“In this study, we present the first evaluation of the highly debated ban on new fossil fuel boilers by assessing its robustness in achieving carbon neutrality under uncertainty, its cost-efficiency, and its distributional effects. First, the ban shifts the strategy for gas resource allocation from gas boilers to a combination of peaking power plants and heat pumps. This new allocation leads to an energy system that both reduces the need for primary energy generation and optimizes utilization of electricity capacities. Second, we demonstrate that achieving carbon neutrality in the residential sector is highly uncertain under the current policy regime. In contrast, we show that the ban is a more robust strategy for achieving climate neutrality, showing no adverse effect on the electricity system while hedging against the lower-than-expected biogas potential. Third, despite costly investments in heating system, the ban leads to lower total system costs over a large range of plausible futures. Fourth, we show that the distributional impacts are highly sensitive to the subsidy design for heat pumps and needs to account for both income and occupation status. When coupled with the French existing subsidy framework, it mitigates vertical inequalities among owner-occupied households but does not extend to those in privately rented homes.”

The repository includes detailed instructions for installation, setup, and running simulations, ensuring reproducibility and ease of use. The methodology involves creating scenarios and running scripts to analyze policy impacts on energy systems. I couldn't manage to install and run the code due to technical problems occurred in personal PC but the repository appears to be functional and well-documented.

We thank the reviewer for their thorough examination of our model's repository. Indeed, the code is primarily designed to run on Unix, which is the standard for remote servers, and has not been specifically optimized for Windows environments. To address this, we have added a warning in the code documentation indicating that, at this time, the code should be run on Unix.

REVIEWERS' COMMENTS

Reviewer #1 (Remarks to the Author):

The authors thoroughly revised the manuscript following the comments and have significantly improved the manuscript. The focus and research questions are now much clearer and the main shortcomings have been addressed. I therefore support the publication of the manuscript.

Reviewer #2 (Remarks to the Author):

Previous comments have been addressed in the revision. The work provides critical insights into the topic.